# Design of an Integrated Controller for a Sweeping Mechanism of a Low-Dust Almond Pickup Machine

**DOI:** 10.3390/s23042046

**Published:** 2023-02-11

**Authors:** Reza Serajian, Jian-Qiao Sun, Reza Ehsani

**Affiliations:** Department of Mechanical Engineering, The University of California Merced, 5200 N. Lake Road, Merced, CA 95343, USA

**Keywords:** feedback control, optimize, stabilize, angular velocity, synchronize, height controller, harvester, dust generation

## Abstract

California is the world’s biggest producer and exporter of almonds. Currently, the sweeping of almonds during the harvest creates a significant amount of dust, causing air pollution in the neighboring urban areas. A low-dust sweeping system was designed to reduce the dust during the sweeping of almonds in the orchard. The system includes a feedback control system to control the sweeper brushes’ height and their angular velocity by adjusting the forward velocity of the harvester and the brushes’ rotational speeds to avoid any extra overlapping sweeping, which increases dust generation. The governing kinematic equations for sweepers’ angular velocity and vehicle forward speed were derived. The feedback controllers for synchronizing these speeds were designed to optimize brush/dust contact to minimize dust generation. The sweepers’ height controller was also designed to stabilize the gap between the brushes and the orchard floor and track the road trajectory. Controllers were simulated and tuned for a fast response for agricultural applications with less than a second response delay. Results showed that the designed system has acceptable performance and generates low amounts of dust within the acceptable range of California ambient air quality standards.

## 1. Introduction

Trunk-shaker harvesting machines are a common tool used in the harvest of almonds in California. These machines function by vibrating the trees, causing the almond nuts to fall to the ground. Once the nuts have reached the desired moisture content, they are gathered into windrows using sweepers. The sweeping process begins by blowing the nuts between the tree lines and then sweeping the almonds toward the middle rows. After the sweeping operation, a pickup machine can then collect the nuts from the ground. One way to decrease the amount of dust generated during the harvest process is to design a more efficient sweeping system that has minimal interaction with the soil. By reducing the amount of soil disturbance during the sweeping process, it is possible to decrease the amount of dust generated, making the harvest process more efficient and less harmful to the environment.

### 1.1. Background

The issue of dust generated during the almond harvesting process has been a significant concern for the manufacturers of the equipment used in the process. To address this issue, several attempts have been made to modify the equipment to reduce the amount of dust produced. These efforts have primarily focused on trying to lower the dust after the pickup step. However, despite these attempts, the dust level generated during the almond harvest process remains high.

Table 1 provides a summary of the modifications made by some of the commercial harvesting equipment manufacturers to reduce dust. One manufacturer, “Exact Corporations” [1], attempted to reduce dust by spraying water on it, but this approach resulted in additional problems, such as the need for maintenance to remove the mud generated in the container and slowing down the harvest process due to the need for refilling the water tank. Another manufacturer, “Flory Industries” [2], tried to reduce dust by making the conveyor belt transfer almonds to the hopper longer, but this approach was not successful in significantly reducing the dust level. The “Jackrabbit Corporation” [3] designed a disk-based cleaning section to separate dust from almonds, but even this approach has its limitations.

It would be more effective if the harvesting system was designed to generate less dust in the first place, rather than focusing on suppressing the dust that has already been generated. Despite the efforts made by various manufacturers such as Weiss-McNair [4], there remains a need to effectively reduce the dust level during the almond harvest process.

### 1.2. Related Works

Previously reported agricultural-related research works in this area have been completed mainly on header-height control systems for the combines. In 2014, Yangmin Xie et al. designed a feedback controller for achieving robust stability and performance for grain combine-harvester header-height control. An offline tracking system was used in their study to determine the shape of the ground before harvesting. In their study, they observed a large elevation change in a short distance due to the rapid changes in ground topology. To avoid causing equipment damage, they limited the maximum combined speed to 0.4 mph (0.64 km/h) using a controller [5]. Shuqi Shang (2022) presented an adaptive method for controlling headers’ height. The height error of cutting stubble is about 2 cm, which meets the requirements of a 5–11 km/h harvesting speed in plain areas [6]. Yangmin Xie (2010) developed a state-feedback LQR controller for grain combine harvesters that allows adjustability according to ground shape [7]. A soybean harvester header-height adjustment system was developed by Chengqian Jin et al. in 2021. They designed it to eliminate the problems of improper harvesting and soil shoveling resulting from low-positioned soybean plants and inadequate header height control. The effect of soil compactness on the ground profile control accuracy of the header-height control system was one of the limitations of their study [8]. Xie et al. (2013) investigated the fundamental performance limitations of their header height control system for a grain combine harvester. In their study, the closed-loop performance was limited by the hydraulic actuator’s time delay. For the system to have a low natural frequency, they had to redesign key parameters, such as suspension elements [9]. The grain combine-harvester header-height tracking controller designed by Al Sawafi et al. (2021) used a fuzzy adaptive PID controller. When the fuzzy system was present, the PID control system changed its gain instantly to compensate. While the combine harvester was moving and tracking the change in the ground topology, the sensor and fuzzy logic were read and recorded. Due to their nonlinear nature, fuzzy controllers are more difficult to set than proportional-integral-derivative controllers, and the fuzzy control’s inputs caused errors in their system [10]. Punit Tulpule et al. (2014) improved header-height control performance in a combine harvester by applying a sensitivity-based integrated robust optimal design methodology. The combination of their design and sequential methods resulted in a more robust and well-integrated system that can be controlled with less control power. Therefore, the vehicle’s speed is increased, and operating costs are reduced [11]. For sugarcane harvester base-cutter control, Jun Zhang et al. (2022) used ground-penetrating radar signals to automatically control the height of the base-cutter; thus, they presented a method based on ground penetration radar (GPR). Using a knockdown roller mounted on a 1.6 GHz GPR antenna, they measured the vertical distance between the ground and air-coupled GPR antenna. Due to limitations caused by a minimum error of 1.76 cm in ground layer detection the base-cutter height only varied by 1.46 cm [12]. A robust two-DOF controller for delayed LTI systems was introduced by Xie et al. (2013). During rapid changes in external signals, such as reference signals and/or disturbances, the system’s feedback control loop is restricted, resulting in a reduced response capability. As a result, the delay term of the feedback controller was eliminated without affecting the H∞, norm of the objective functions [13]. One-rotor orchard inter-row analysis based on tine trajectory was designed by Lei XiaoHui et al. in 2019. An inter-row rake with one rotor was developed to improve orchard mechanization efficiency [14].

### 1.3. Objectives

This study aimed to design a height and speed controller for a newly designed almond sweeper and pickup system. We designed PID feedback controllers using SIMULINK and identified the system dynamics of the mechanism. A minimum amount of contact with the orchard floor was achieved by synchronizing the brush rotational speed with the forward velocity of the harvester while adjusting the sweepers’ heights to minimize dust generation by ensuring they only touched nuts, not dirt, debris, stones or fruit.

## 2. Materials and Methods

### 2.1. Mathematical Modeling for Sweeping System

Figure 1 shows the prototype of the newly designed sweeper system. During the sweeping process (Figure 2), in the machine’s operation, it moves along the *Y*-axis while the disc rotates in a counterclockwise direction. The negative direction of the *X*-axis is the location where the almonds are placed on the ground. As the disc rotates, the tines are utilized to sweep the nuts into a row. During the rotation of the disc from point 1 to point 2, the tines are in contact with the nuts, guiding them into a row. However, during the rotation from point 2 back to point 1, there is no contact between the tines and the ground.

The forward speed of the sweeper was set at v, the rotational speed was ω, the rotational radius was r, the arm number was n, and the tine working width was L. The plane trajectory of the movement of the tines is illustrated in Figure 3. The diagram shows the outermost and innermost positions of the tines on both arms of the device. Specifically, point a and point a’ represent the outermost positions of the tines that are working on the first arm. Similarly, point b and point b’ represent the innermost positions of the tines that are operating on the first arm. On the second arm, point c and point c’ represent the outermost positions of the tines, while point d and point d’ indicate the innermost positions of the tines. This diagram helps to clearly visualize the range of motion of the tines and their positions on the device’s arms.

The thick solid line area, designated as (aa’b’b), represents the trajectory of the first arm’s tines, while the thick dashed line area, marked as (cc’d’d), signifies the path of the second arm’s tines. The uncrossed area of the tines during their operation is referred to as the “miss-raking area”. On the other hand, the “repeat-raking area” is the portion of the working area where the tines have crossed two or more times. To quantify the performance of the tine operation, the miss-raking rate is defined as the ratio of the miss-raking area to the total working area in one cycle of disc rotation, while the repeat-raking rate is determined as the ratio of the repeat-raking area to the working area in one cycle of disc rotation.

The movement equation of point a is:(1)$$\{\begin{array}{c} X_{a}=r\cos\omega t\\ Y_{a}=r\sin\omega t+vt \end{array}$$
where *X_a_* = value of point *a* on the X axis;

$Y_{a}$ = value of point *a* on the Y axis;

t = working time.

The movement equation of point d is:(2)$$\{\begin{array}{c} X_{d}=\left(r-l\right)\cos\left(\omega t-\frac{2\pi }{n}\right)\\ Y_{d}=\left(r-l\right)\sin(\omega t-\frac{2\pi }{n})+vt \end{array}$$
where $X_{d}$ = value of point *d* on the X axis;

$Y_{d}$ = value of point *d* on the Y axis.

Calculation of the max. values of point “a” and point “d” on the Y axis by derivative $Y_{a}$ and $Y_{d}$ in Equations (1) and (2). They are:(3)$$Y_{a_{max}}=\left(\frac{v}{\omega }\right)\cos^{-1}(-\frac{v}{\omega r})+r\sin(\cos^{-1}\left(-\frac{v}{\omega r}\right))$$
(4)$$Y_{d_{max}}=v\left(\left(\frac{2\pi }{\omega n}\right)+\left(\frac{1}{\omega }\right)\left(\cos^{-1}\left(-v\omega \left(r-l\right)\right)\right)\right)+\left(r-l\right)\sin(\cos^{-1}\left(-\frac{v}{\omega \left(r-l\right)}\right))$$
where $Y_{a_{max}}$ = max. value of point *a* on the Y axis;

$Y_{d_{max}}$ = max. value of point *d* on the Y axis.

In a single rotation cycle of the brush, if the maximum height of Y for point “a” ($Y_{a_{max}}$) is less than the maximum height of Y for point “d” ($Y_{d_{max}}$), there will be two points of intersection between the trajectory of point “*a*” and the trajectory of point “*d*”. The area located between these two intersections is referred to as the “miss-raking zone”. This miss-raking zone represents a portion of the surface that is not effectively cleaned or raked by the brush. It is important to take the position of the miss-raking zone into consideration when using a brush to ensure a thorough cleaning process.

If $Y_{a_{max}}$ = $Y_{d_{max}}$, there is only one tangent point between point “*a*” trajectory and point “*d*” trajectory, and $X_{a_{max}}$ = $X_{d_{max}}$ at this moment, and the repeat-raking zone is the least, this is the optimal working state. If $Y_{a_{max}}$ > $Y_{d_{max}}$, there would be a larger repeat-raking zone, a faster movement speed, and a lower disc rotational speed, which can lead to improved brush performance. The ideal working condition for the brush is achieved when there is no gap or minimal overlap between the adjacent working areas of the tines. This ensures that all the area in front of the brush is effectively swept and that the tines are functioning optimally; therefore, this means $Y_{a_{max}}$ − $Y_{d_{max}}$ ≥ 0. Based on the detailed parameters, we obtained:(5)$$\sqrt{1-\left(\frac{v^{2}}{\omega^{2}r^{2}}\right)}-\sqrt{{\left(1-\frac{l}{r}\right)}^{2}-\left(\frac{v^{2}}{\omega^{2}r^{2}}\right)}+\frac{v}{\omega r}\left[\cos^{-1}\frac{v}{\omega \left(r-l\right)}-\cos^{-1}\frac{v}{\omega r}\right]-\frac{2\pi v}{n\omega r}\geq 0$$

Equation (5) represents the optimal kinematic model for a one-rotor horizontal brush. In order for the brush to effectively sweep all the area in front of it, this equation should equal zero. This indicates that the brush is executing the proper motion and achieving complete coverage of the area in front of it.

### 2.2. Sweeper Angular Velocity Controller Design

The feed-forward reference adjustment for our control system is a critical component, as it helps to ensure accurate and efficient performance. The adjustment is based on a mathematical relationship between the forward speed and angular velocity of the brushes, which is described in Equation (5). Using the data obtained from the forward speed magnetic sensor, we can determine the precise angular velocity required for the brushes to operate effectively. In other words, the feed-forward reference adjustment allows us to adjust the speed and movement of the brushes based on the forward velocity of the harvester, which is a crucial factor in determining the performance of the system. The combination of this reference adjustment with the data from the magnetic sensor ensures that the brushes operate at optimal speed and efficiency, contributing to the overall success of our control system.

#### 2.2.1. Sweeper Motor System Identification

The brushes in question are equipped with a Zipo electric motor, identified by the model number ATP50. This motor is responsible for powering the brush, and its output torque was boosted using chains and sprockets, which connect the motor shaft to the brush shaft. The electrical motors used in this study were analyzed, and their dynamics were determined through experimentation and the use of the MATLAB System Identification package. Unfortunately, the transfer functions for these motors were not readily available, so an experimental approach was taken to understand the electrical motors and brush dynamics. To accurately measure the performance of the sweeping system, two different instruments were employed. The first was a tachometer, which was used to determine the forward speed of the system. Additionally, a spindle speed meter sensor was used to measure the rotational velocity of the sweeping brush. The sweeping motor had a limited maximum angular velocity of 65 RPM, which was a key factor in determining the maximum forward speed. According to the results of Equation (5), this maximum forward speed was limited to 10 m/s. To further analyze the system, a graph was created showing the relationship between brush angular velocity and motor input voltage, as depicted in Figure 4. To gain a deeper understanding of the system’s dynamics, the MATLAB System Identification Toolbox was used to conduct a detailed analysis.

The motor and brush transfer function is as follows:(6)$$B_{S}\left(S\right)=\frac{\theta }{V}=\frac{8.7208}{0.0475\;S^{2}+0.1173\;S+1}$$

#### 2.2.2. Angular Velocity Control System

The block diagram in Figure 5 served as the basis for the construction of a PID (Proportional Integral Derivative) controller for the dynamic system. By studying the block diagram, it was possible to design and implement a PID controller that would effectively control the behavior of the system. This type of controller is commonly used in a variety of applications and is well-suited to this particular dynamic system due to its ability to accurately respond to changes in system behavior and maintain control over the system. The implementation of the PID controller in accordance with the block diagram in Figure 5 has allowed for precise control over the dynamic system and ensured that the system operates in a stable and efficient manner.

The diagram represents the working principle of our control system, which is designed to regulate brush angular velocity in an almond harvesting system. The system starts with a feed-forward section, which tunes the brush angular velocity based on the forward speed of the vehicle. The forward speed is manually input into the system and the brush angular velocity is determined based on this input, which is then fed back into the feedback system for regulation. A simulation was performed using a function for vehicle forward speed as the input to the control system. This simulation allowed us to determine the desired versus measured omega (brush angular velocity) and the necessary voltage for the motor to apply the desired angular velocity. The results of the simulation are shown in Figure 6, which highlights the performance of the control system and its ability to accurately regulate brush angular velocity.

In summary, the diagram illustrates the functioning of a control system that is designed to regulate brush angular velocity in an almond harvesting system. The feed-forward section tunes the brush angular velocity based on the vehicle’s forward speed, and the simulation results show the accuracy and efficiency of the system in regulating brush angular velocity.

Figure 6 demonstrates the performance of the control system in regulating brush angular velocity based on the forward speed of the vehicle. As the forward speed increases, the angular velocity also increases, following the trend defined in the first section of the omega-time curve. This relationship is confirmed by Equation (5), which demonstrates the correlation between forward speed and brush angular speed.

In the second section of the curve, we see a constant forward speed that is maintained by a fixed angular velocity. The vehicle moves forward with two decelerations and two constant speeds. Figure 7 shows that the desired omega tracks the measured omega with good accuracy. This indicates that the control system is functioning effectively in regulating the brush angular velocity based on the forward speed of the vehicle.

The settling time, which is the time it takes for the system to reach its final steady-state value, was found to be 0.57 s, which is a small value and indicates that the control system responds quickly to changes in the forward speed. Additionally, the overshoot, which is the maximum deviation from the final steady-state value, is also minimal, further demonstrating the effectiveness of the control system.

#### 2.2.3. Linear Actuator Height Controller Design

For each sweeper, we must use a closed-loop control system to specify the brush height so that the gap with the ground is 0.5 inches. This 0.5-inch distance between the brush edge and floor is less than the thickness of one almond with shell on the orchard floor, which will help to just sweep the fruit and minimize brush/orchard floor contact to further minimize dust generation.

#### 2.2.4. Linear Actuator System Identification

The height of each brush is controlled through the use of an actuator. The specific actuators utilized in this study were manufactured by Progressive Automation Co. and are identified by the part number PA-04-6-100. These actuators have a 12 VDC power supply, a 6-inch stroke, and a maximum weight capacity of 100 lbs. To ensure precise control over the length of the actuators, feedback from two optical distance sensors is employed. These sensors are mounted on each sweeper and provide real-time data to inform the control system and make necessary adjustments to the actuator length.

To determine the average speed of the actuator’s motion, an experimental method was used where the weight was lifted down at a faster rate than it was lifted up. However, this approach introduced some inaccuracies in the calculations, which could be corrected through compensation in the control simulation to ensure that the lifting speed is similar in both the upward and downward motions. The dynamic behavior of the actuator/brush height system was also analyzed using the MATLAB System Identification Package, providing a deeper understanding of the system’s performance.

The transfer function of the height actuator:(7)$$A_{S}\left(S\right)=\frac{L}{V}=\frac{0.0078156}{0.21617\;\;S^{2}+0.1951\;S+1}$$

#### 2.2.5. Linear Actuator Control System

The adjustment compensator for the motor’s lift up/down speeds:(8)$$P+I\left(\frac{1}{S}\right)+D\left(\frac{N}{1+N\frac{1}{S}}\right)$$

The compensator uses the following parameters:



P=20,





I=0.05,





D=10,



The number of the filter coefficients (N)=10

When designing the control system for a sweeping system with two brushes, it is essential to consider the unique requirements of each brush. This is due to the fact that in a real-world application, the controller must accurately position different tracks in front of each brush, ensuring that no fruit is missed and that the brush is not damaged in the process. To achieve this goal, it is necessary to create two separate control block diagrams. These diagrams will outline the specific requirements and parameters for each brush, allowing for precise and effective control over the sweeping system. In this way, we can ensure that the sweeping system operates efficiently and effectively, minimizing any damage to the brushes and maximizing the collection of fruit.

As a semi-realistic track with a period of 35 s, we have established a sinusoidal wave to symbolize the height controller located in front of both brush units. The height controller plays a crucial role in ensuring that the brushes maintain a consistent height as they travel along the track, which is essential for creating a smooth and even surface. The use of a sinusoidal wave to represent the height controller allows us to accurately simulate the changing heights of the brushes in a controlled and predictable manner. This helps us to thoroughly test the performance of the brushes and ensure that they will work effectively in real-world conditions. Overall, the sinusoidal wave serves as an effective tool for modeling and evaluating the height control system of the brushes.

The left graph in Figure 8 provides a visual representation of the response of the front brush height control, as well as the voltage that is applied to it. As the actuator voltage graph indicates, the voltage remains constant at 12 volts from time 0 to 5 s and then changes to −12 volts from time 12 to 18 s. It is important to note that our actuators have a maximum capacity of 12 volts, so they cannot exceed their rated voltage when more power is needed. In the right graph in Figure 8, we can observe the impact of saturation on the actuator response time, which results in a decrease in the lift up/down speed during the period of saturation. The slow response time of the actuator is due to the limitations imposed by saturation, which negatively affects the overall performance of the system. To overcome this issue, engineers and designers need to consider ways to prevent saturation from occurring, so that the actuators can operate at their maximum potential, providing fast and reliable control over the front brush height. The control system block diagram for two front brushes is represented in Figure 9.

## 3. Performance Results

Our lab test was thoroughly designed to mimic real orchard conditions, and we made sure to use soil, dried leaves, debris, etc., that were representative of an actual almond orchard. We used low-dust harvester results as a baseline to compare our results against and were able to demonstrate that our newly designed almond sweeper and pickup system generated significantly less dust (measured as $PM_{2.5}$ and $PM_{10}$) compared to the existing low-dust harvester being used in the field [15]. We recorded these results and compared them to data from tests performed in real orchards. Table 2 showcases the comparison between our lab test results and the real orchard tests. Despite the promising results from our lab test, we understand that it is still important to perform tests in real orchards to obtain more data and optimize our design further. Hence, we plan to conduct tests during the upcoming harvest season to validate our design under real orchard conditions.

## 4. Conclusions

The primary objective of this study is to develop a control system for the sweepers of an almond harvester that can maintain optimum contact between the brush, orchard floor, and fruit. This is performed to reduce the amount of dust generated during the harvest process, which is a major concern during the California harvest season. Dust can cause respiratory problems for the workers involved in the harvest as well as causing damage to the quality of the fruit. In order to make our control system compatible with the existing system, two feedback systems were designed and tested. Although there were a few limitations in our system, including the limited rated voltage of the actuators that resulted in delayed response time and the limited ability of the electrical motors to increase Omega above 65 rpm, the controllers still performed well enough for our harvester mechanism. Despite these limitations, our study aims to address a critical problem in the almond harvest industry and provide a solution that can be implemented to reduce dust generation during the harvest process.

## Figures and Tables

**Figure 1 sensors-23-02046-f001:**
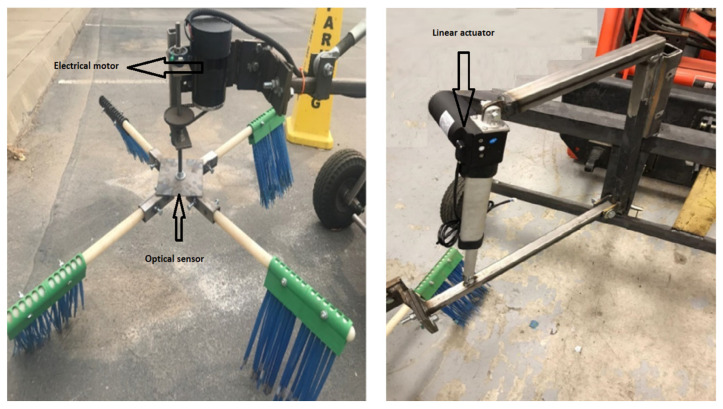
Sweeping mechanism.

**Figure 2 sensors-23-02046-f002:**
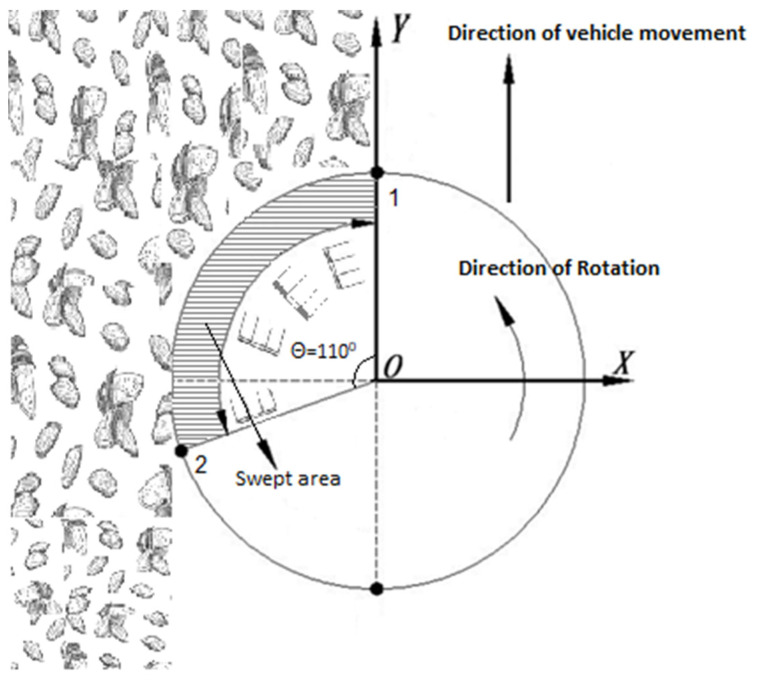
The stationary plane trajectory of the sweeper.

**Figure 3 sensors-23-02046-f003:**
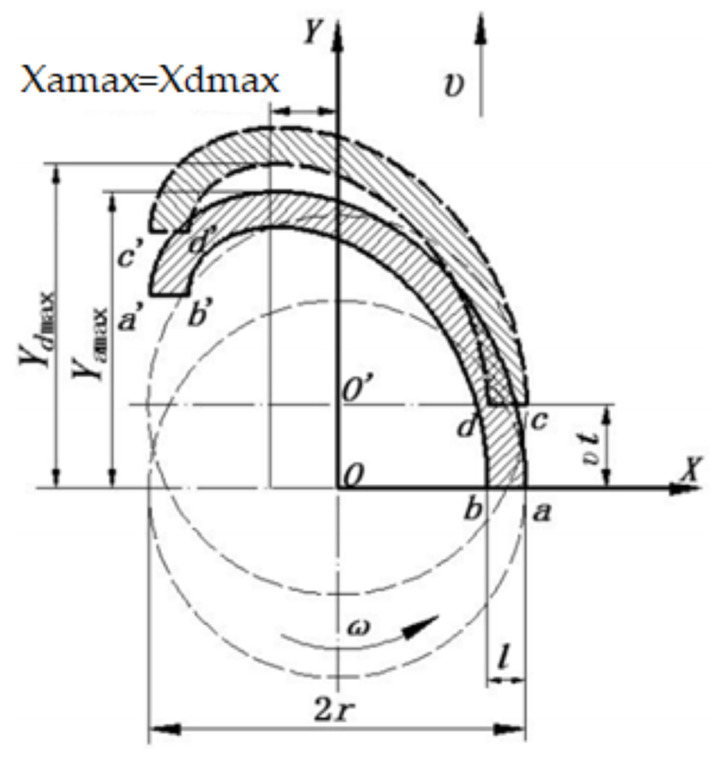
The plane trajectory of tines’ movement.

**Figure 4 sensors-23-02046-f004:**
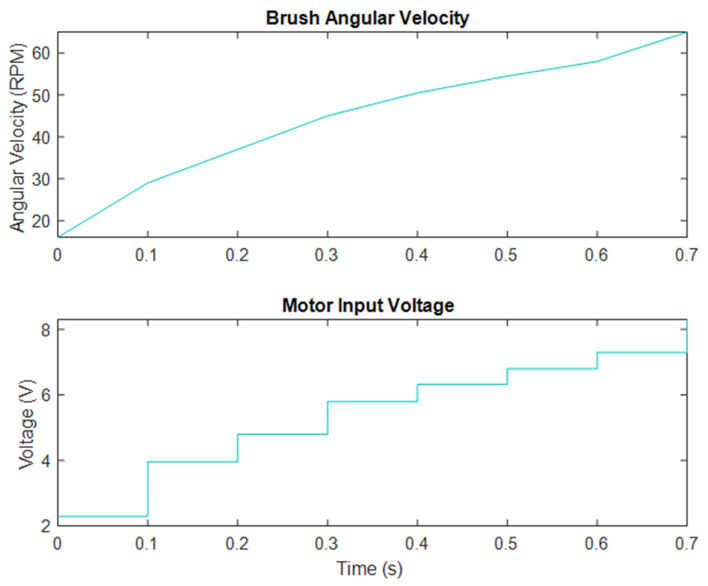
Brush angular velocity and motor input voltage.

**Figure 5 sensors-23-02046-f005:**
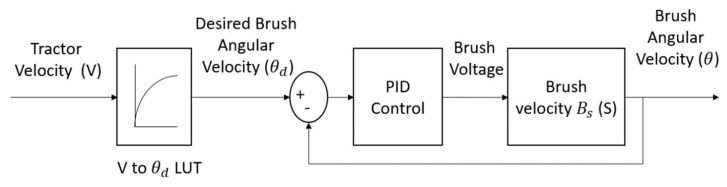
Control system block diagram for front brush angular speed regulation.

**Figure 6 sensors-23-02046-f006:**
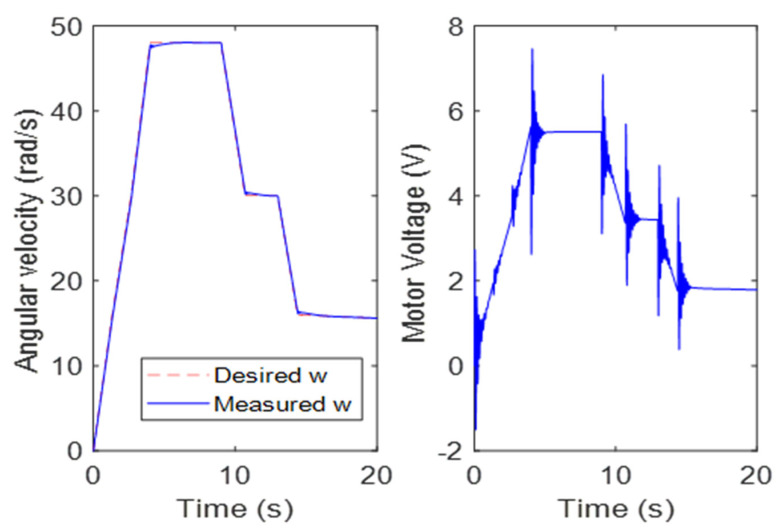
Front brush tracking control response with PID controller.

**Figure 7 sensors-23-02046-f007:**
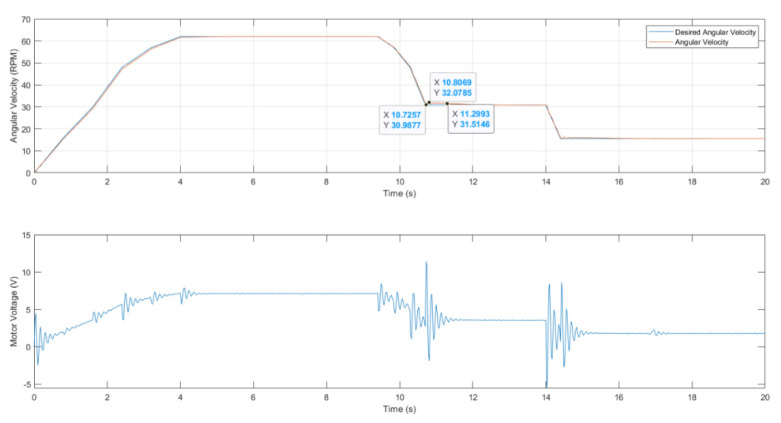
Front brush-tracking control response with a PID controller with overshoot and settling time.

**Figure 8 sensors-23-02046-f008:**
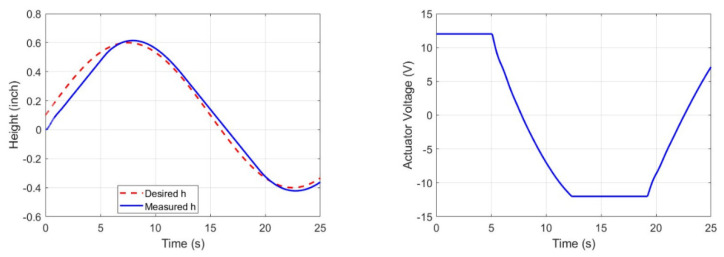
Front brush height control response with a PID controller.

**Figure 9 sensors-23-02046-f009:**
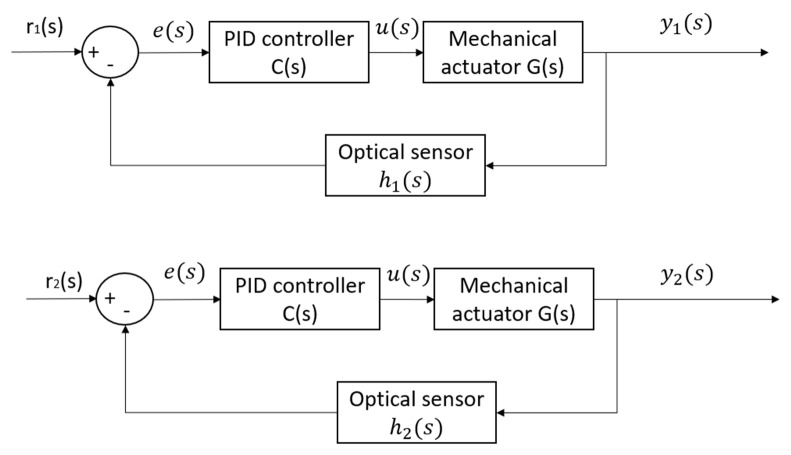
Control system block diagram for front brushes’ height.

**Table 1 sensors-23-02046-t001:** Most recent commercial low-dust almond harvester.

Manufacturer	Model	Drive	Technology Notes
**Harvest pickup equipment**			
Exact Corporation (Modesto, CA, USA)	E-3800	Pull-behind PTO	Features a water misting and brush system at the separation fan. Reduced discharge airspeed
	E-4000	Pull-behind PTO	
	E-7000SP		
Flory Industries (Salida, CA, USA)	860 XL	Pull-behind PTO	Reduce fan speed, longer cleaning chain length, and changes to the location of dust discharge. Reduced discharge airspeed
	8600 XL	Self-Propelled	
	8770 XL	Self-Propelled	
Jackrabbit (Ripon, CA, USA)	Harvester	Pull-behind PTO	Disk-based cleaning section, With twin-rod out load chain. Adjustable fan speed and damper
Weiss-McNair (Chico, CA, USA)	9800 California Special	Pull-behind PTO	Reduced fan speed, fan location, enlarged vacuum and separation chambers, and cleaning chain design. It reduced discharge airspeed
	Magnum X	Self-Propelled	
**Shaker/Sweeper combination unit**			
Tenias (Turlock, CA, USA)	Almond Harvester	Self-Propelled	Shaker drops nuts onto a plate and funnels them into windrows. Eliminates the need for the sweeping process(sweeper/shaker in one combined unit)

**Table 2 sensors-23-02046-t002:** Dust generation for the lab test versus field test results.

Average $PM_{2.5}$ (Lab Test)	Average $PM_{2.5}$ (Field Test)	Improvement	Average $PM_{10}$ (Lab Test)	Average $PM_{10}$ (Field Test)	Improvement
44 $\frac{\mu g}{m^{3}}$	700 $\frac{\mu g}{m^{3}}$	94%	80 $\frac{\mu g}{m^{3}}$	1015 $\frac{\mu g}{m^{3}}$	92%

## Data Availability

Data is not available at this moment due to no field test being done.

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
