# Peer review of "Design of an Integrated Controller for a Sweeping Mechanism of a Low-Dust Almond Pickup Machine"

_sensors, 2023, doi:10.3390/s23042046_

Round 1

Reviewer 1 Report

Accept.

Reviewer 2 Report

This paper presents a design of an integrated controller for a sweeping mechanism of a low-dust almond pickup machine. The topic is certainly worthy of investigation nowadays, but the manuscript suffers from the following issues:

·         The research question and has not been put forward clearly.

·         Abstract should be more focused to be a standalone summary of your paper. Where it highlights key content areas, your research purpose, the relevance or importance of your work, and the main outcomes. Should be revised!

·         Section 1 (Introduction) is too long, so, it should be separated from the literature review (Section 2 Related Study).

·         A comparison table between the proposed system and other related studies is highly recommended to be added in Section 2 (2 Related Study).  Then, motivations can be easily drawn.

·         More related studies should be considered in related study section.

·         The proposed mathematical model should be in section; while visual figures could be in a separated section (e.g., Results and Discussion).

·         How the model validates the optimality of the rotational speeds of sweepers as well as the length of actuators?

·         Is there any practical work to be added for more validation?

·         The manuscript would greatly benefit from proofreading, as there are lots of grammar errors, vague statements and claims, and typos.

·         Figures could be improved from size and resolution perspectives.

·         Overall, the manuscript suffers from serious issues, so it is recommended to be a major revision. 

Reviewer 3 Report

It is a very attractive topic and well handled. Because the design of almond machines is very important in agriculture. Nevertheless, it is better to explain the results and discussion section by performing sensitivity analysis of the performance of the designed mechanism. It is also recommended to use test design methods such as RSM.

Round 2

Reviewer 2 Report

Accept in present form